# Municipal Solid Waste Collection: Challenges, Strategies and Perspectives in the Optimization of a Municipal Route in a Southern Mexican Town

Viridiana Del Carmen-Niño [1], Ricardo Herrera-Navarrete [2], Ana Laura Juárez-López [2,*], María Laura Sampedro-Rosas [2] and Maximino Reyes-Umaña [2]

1   School of Sustainable Development, Autonomous University of Guerrero, Carretera Nacional Acapulco-Zihuatanejo km 106+900, Colonia Las Tunas, Tecpan de Galeana 40900, Guerrero, Mexico
2   PhD in Environmental Sciences, Regional Development Sciences Center, Autonomous University of Guerrero, Calle Privada de Laurel No. 13, Colonia El Roble, Acapulco 39640, Guerrero, Mexico
*   Correspondence: aljuarez@uagro.mx

**Abstract:** Solid waste management represents a challenge for municipalities, particularly at the collection stage. The high costs involved in its operation make it difficult to provide the service in all its communities, which results in bad practices (burning, burying, or dumping into a river) and the proliferation of open dumps. Collection efficiency is aggravated by poor route planning, narrow road networks, and irregular scheduling. This research proposes and develops the following objectives: (1) an analysis of the technical and logistical conditions of a town, (2) an improved municipal route for waste collection, and (3) the practical implications identified in the optimization of the service (challenges, strategies, and perspectives). During the study period (2018, 2019 and 2021), the following steps were carried out: (1) field trips to monitor the formal and informal routes, georeferencing each stop with a global positioning system, (2) semi-structured interviews to route personnel to learn about technical and logistical aspects, (3) the downloading of cartographic data for digitization, and (4) a network analysis aimed at designing an optimal route for formal collection and the spatial scope of the informal routes. The current technical and logistical analysis detected inefficiency in collection due to weak municipal operational planning and the irregular frequency of visits to the locality. The locality produces an average of 2.8 tons per day and its largest volume is organic waste (68.3%), followed by non-recyclable inorganic waste (21.2%), and recyclable waste (10.5%). In terms of results for the optimization of the municipal collection route, it is estimated that there will be an improvement in the time of the day of approximately 2 h less, going through 95 points when its schedule is 60 collection points, while the distance factor does not suffer changes due to the fact that there are no alternate routes between the departure and destination route. Among the main challenges faced by the municipalities are the limited budgets for basic sanitation services, which is why technological strategies and trained human resources are required for better municipal solid waste management. From a technical perspective, geographic information systems are a current trend to model and optimize service routes, with which a better scenario can be proposed. From a social perspective, community participation works as a key factor to carry out activities focused on finding solutions to the problems related to municipal solid waste management.

**Keywords:** town; municipal collection; GIS; route optimization; network analysis

## 1. Introduction

The worldwide population growth and production and consumption activities increase the generation of municipal solid waste (MSW), so its collection represents a serious socio-environmental problem [1,2]. In this sense, Saja et al. [3] point out that municipal solid waste management (MSWM) is a challenge for municipal authorities, mainly in developing countries due to the constant increase in waste generation, the high costs of operation, the

lack of understanding of the technicalities involved in the process, and a low level of citizen participation. It is important to note that MSWM is linked to the regulations and actions of each region to achieve environmental benefits and resource optimization [4].

According to United Nations data [5], Latin America and the Caribbean generates approximately 10% of global waste. Although its final disposal has improved in recent decades, approximately 145,000 t/day is still destined for open-air dumps, burning, or other inadequate practices. In Mexico alone, up to the year 2000, 30 million tons were produced; in 2015, the amount rose to 53.1 million, an increase of more than 70% in the span of 15 years [6]. On the other hand, municipal governments present budget limitations, in addition to a lack of reliable information, which limits decision-making and the formulation of stricter public policies [7].

Mexican legislation establishes in Article 115 of the Mexican Constitution that municipalities are responsible for the collection of urban solid waste. The General Law for the Prevention and Integral Management of Waste and the General Law for Ecological Equilibrium and Environmental Protection derive from this law, which establishes actions related to waste management that must be adapted to the conditions and needs of each place and allow compliance with Mexican Official Standards to achieve the objectives of valorization, and sanitary, environmental, technological, economic and social efficiency [8,9].

It is important to note that, in addition to the formal MSWM system, there is an informal system that operates as an essential part of the service, whose operation can be beneficial. Despite being considered a competitor that carries out activities that are not financed, recognized, supported, or organized by the formal authorities in charge of the public service [10], this sector plays an important role in the recovery of waste, which contributes to mitigating the negative environmental impacts [11].

This type of problem becomes more severe the farther away the localities are from the municipal capital. In these areas, which can even go to the extreme of completely lacking the service, open dumps are formed, generally in ravines or land surrounded by agricultural crops; it is also common for waste to be buried in pits made by the community or burned in the open air [8,12]. On the other hand, in the case of the towns that do have a collection service, the operational problems they face are different: narrow roads restrict the possibility of large vehicles passing, and, for this reason, manual operations must be carried out, an activity that increases costs [13].

In this context, it is feasible to use a geographic information system (GIS) as a strategy to optimize collection times and reduce costs on routes. GIS is a powerful tool for handling, storing, managing, and analyzing large spatial data sets from a variety of sources [14]. For example, Gallardo et al. [15] used GIS to design MSW collection routes and generate a municipal management plan. Araiza and José [16] showed, in their study, the importance of spatial analysis for the improvement in the waste collection stage in two localities in Chiapas, Mexico, using GIS. According to Hoke and Yalcinkaya [17], several studies have opted for route optimization through GIS models for optimal routing and container reallocation, as well as non-homogeneous vehicle-routing models for the optimization of MSW collection.

In this regard, technology is a complement to achieve efficient MSW management. Therefore, this research analyses the technical and logistical conditions of a town and proposes an improved municipal route for waste collection while identifying the practical implications in the optimization of the service (challenges, strategies and perspectives). Although the issue of optimizing waste collection routes in urban complexes has already been studied, its analysis is considered relevant in contexts far from municipal centers, where open-air dumps commonly proliferate. While other studies present scenarios of the best alternative route, this one seeks to improve service times due to the lack of alternative roads.

While it must be recognized that a high number of years of accumulated experience and knowledge produces benefits in MSWM, it is also important to emphasize that there is still much to be done to increase the performance of the system and reduce costs [17].

However, there is a need at the national level to compile the existing information of this type and produce new data, but a major gap in MSWM in Mexico is the lack of reliable information [7]. This research is of great relevance because the method allowed us to analyze, from a spatial perspective, the technical and logistical conditions of both formal and informal routes, which could help to improve MSWM, mainly in localities far from the municipal capital, where the collection service is not very efficient due to unreliable municipal management.

## 2. Materials and Methods

This research has a mixed design (quantitative and qualitative) and includes a spatial analysis and a series of interviews. To carry out the spatial analysis, the following supplies were used: topographic charts from the database of the National Institute of Statistics and Geography in Mexico [18], value-added data from the Open Spatial Data Infrastructure [19], reference points from a global positioning system (GPS), and the network analysis tool of the ArcGIS software version 10.5. It should be noted that this tool has been useful for the optimization of transportation routes and enables the generation of optimal scenarios. ArcGIS has proven to be effective, easy to use, and time-saving in the field of traffic engineering and transportation planning [20]. In addition, network analysis quantifies accessibility, flow, or efficiency to predict multiple phenomena, such as transportation behavior, land-use change and health [21].

### 2.1. Description of the Study Area

The research was conducted in Xaltianguis, a town located about forty kilometers away from its municipal administrative capital, Acapulco de Juárez, Guerrero (Figure 1). It is classified as urban because of its population size of 6564 inhabitants [22]. It is located northwest of Acapulco between parallels 17°05′58″ north latitude and −99°42′51″ west longitude of the prime meridian and at an altitude of 533 m above sea level [22]. It is considered a locality with a low social backwardness index, which measures four main indicators: educational backwardness, access to health services, access to basic services in housing, and housing quality and space [23].

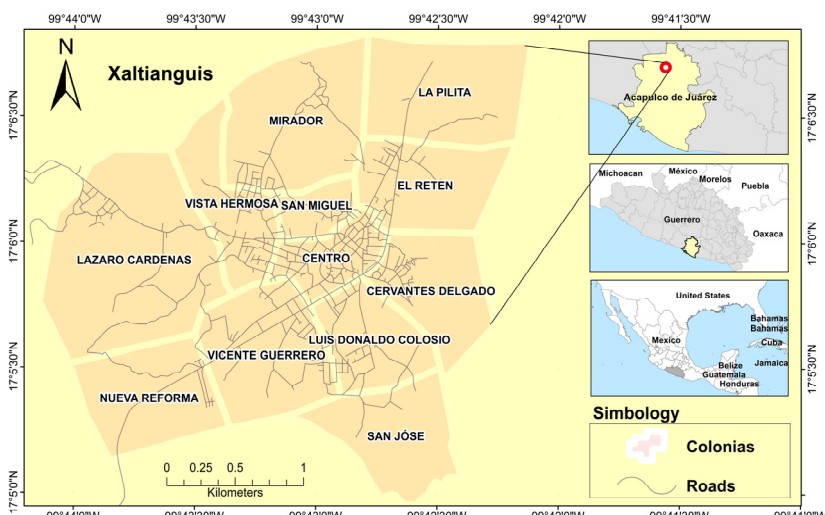

**Figure 1.** Location of the study area divided into twelve colonias.

According to Del Carmen-Niño et al. [24], the municipality is made up of twelve colonias: Centro, Cervantes Delgado, El Retén, El Mirador, Lázaro Cárdenas, Luis Donaldo Colosio, Nueva Reforma, Vicente Guerrero, Vista Hermosa, La Pilita, San José and San Miguel (Figure 1). It is important to mention that the waste collection service depends on Acapulco, which is a popular beach resort with high and low tourist seasons. Due to the city's economic dependence on this activity, there is a policy of prioritizing the provision

of public services to visitors, although this situation may lead to a decrease in attention to certain areas of the municipality [25].

### 2.2. Data Collection

The first step was to conduct several field trips during 2018 and 2019 along the formal and informal routes to identify the collection points, calculate the number of stops and estimate the service time. Next, between November and December 2021, the collection points of the formal route were updated through a global positioning system (GPS Garmin Etrex 10). At the same time, semi-structured interviews were applied to both informal and municipal vehicle operators to learn about technical and logistical aspects. A total of 142 collection points were identified during 2018 and 2019 surveys, and 270 collection points were identified in 2021. The optimization was performed on the latter points (Figure 2). Subsequently, topographic information was obtained from the INEGI platform, which contains the coordinates of the study area, points of interest, and main and secondary roads of the locality in Shapefile format [18]. The field information and the digitalization of the topographic information allowed us to obtain vector data, which formed the database, and was the main input for the network analysis.

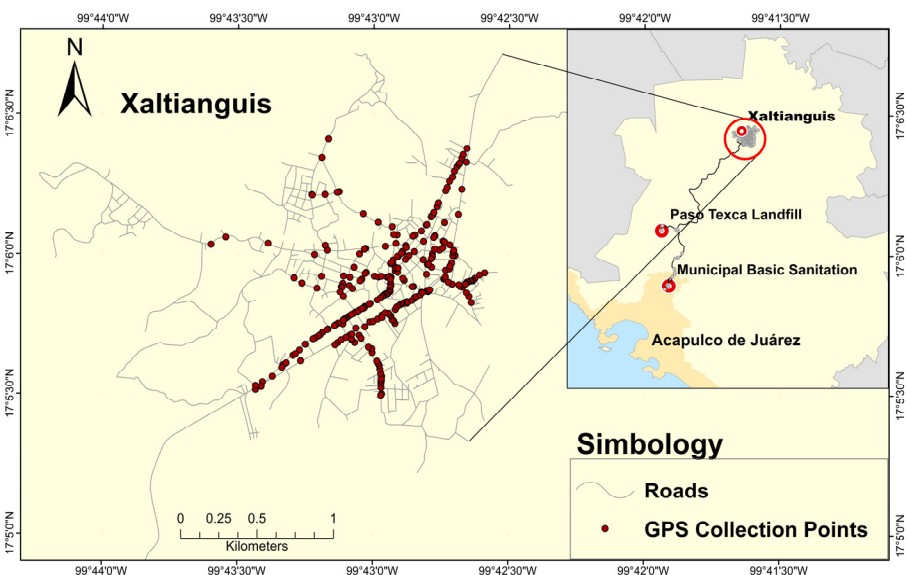

**Figure 2.** Waste collection points obtained by GPS.

### 2.3. Network Analysis Using a GIS

GIS technology has been widely adopted as a promising approach to automate the planning process in waste management, as it can contain spatial attribute data that allow digitizing the object and its attributes effectively with geographic accuracy [26]. In this context, ArcGIS with its network analysis extension is a GIS that allows users to dynamically model realistic network conditions, including turning restrictions, speed limits, and traffic conditions [20].

Network analysis is a complex process that must combine social, environmental, and technical aspects [27], as well as consider the travel restrictions and geographical conditions in some areas. This important task involves processing a large amount of geospatial data that, in some cases, must be digitized for proper operation. To carry out this network analysis, the attributes considered by Yachai et al. [28] were taken into account, adding others that were considered essential for the process:

- Impedance. The measurement, which may include distance, time, or speed of multiple trips per distance. The optimal path should be the one with the lowest impedance, which results in the lowest cost route in time or distance.

- Capacity. The maximum number of people or units that the facility can attend, contain, or assign.
- Solid waste departure and destination. The location of the starting point and destination of the solid wastes will allow a more defined route to be drawn and take into account the times of attention for their departure.
- Start and end of day. It allows the programming of the first visit after the start time established for the activity; however, the end of daytime does not represent a restriction to continue the activity.

### 2.3.1. Create Network Dataset

The dataset was generated from the road network, which contains the links, pathways, and junction points. The network dataset was created with geodatabase in ArcCatalog. The distance of the road network for this analysis is 69.50 km. The road network considers the location of the town, final disposal site, and depot where the units are kept overnight and from where they depart (Figure 3). All data were projected in Universal Transverse Mercator (UTM) zone 14N using the Web Mercator Coordinate System, WGS 1984.

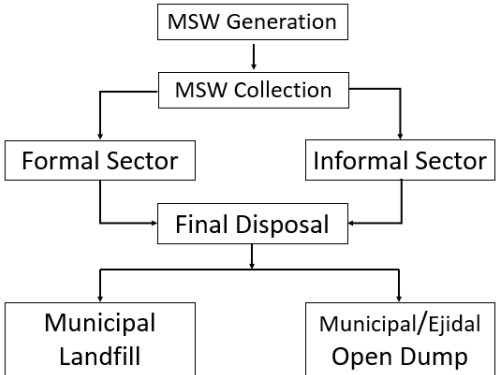

**Figure 3.** Solid waste management system: Xaltianguis operation model. Source: own elaboration.

### 2.3.2. GIS constraints and analysis

The software has the capability to determine constraints through barriers along the path. However, this study excluded critical constraints to the waste collection routes; it only considered the following criteria:

1. All roads are two-way. However, some streets in the communities are difficult to access.
2. The travel speed is slower in the community, ranging between 40 and 60 km/h.
3. The official final disposal site (landfill) has a slow internal access (dirt road); therefore, entry and unloading times take approximately 45 min and are considered in the route times.

## 3. Results

### 3.1. Waste Management System

Figure 3 shows the operation of MSW management in Xaltianguis. It is restricted to the collection and disposal of the waste generated; a simple management system, usually applied at the locality level in accordance with Mexican regulations [9]. Other management practices for waste disposal were also identified, such as waste burning, the collection of recyclable inorganic waste, and composting of organic waste, as well as the potential for population participation in waste management. It is estimated that Xaltianguis generates 2.8 t/d of waste, of which 68.3% is organic and the rest is inorganic [29].

It is important to note that in this town, there is an Ejidal open dump (EOD) where waste is disposed of by an informal collection system and by some inhabitants without any type of environmental control or regulation. In addition, there are other open dumps that

operate clandestinely, (located on the banks of the Xaltianguis River. This river runs across the town and supplies the population with drinking water. There are also open dumps in ravines or on land surrounded by agricultural crops, where the intentional burning of waste by the population persists, generating pollution and affecting human health.

### 3.2. Collection Service

3.2.1. Formal Service (Municipal)

In 2018 and 2019, the provision of a formal service for the collection of solid waste in the locality was carried out by municipal basic sanitation management (BSM). This service performs mixed collections of solid waste, that is, without any separation. The operation of this formal route combined the methods of fixed stops on secondary streets and sidewalks on main streets. Another important point is that the arrival of tourists during the holiday seasons also affects the service, given that, under these conditions, they can be removed from the locality for two or even three weeks to cover the tourist area.

On the other hand, the assignment of trucks for the routes does not consider the terrain conditions or type of vehicle. A side-loading truck without compaction with a capacity of six tons (t) is used on a single established route. The staff consists of a driver and one or two helpers who are tipped USD 0.25–0.50 by families. Waste collection rarely took place in the mornings; it usually began at 2:00 PM when most of the educational and health institutions had already closed their doors, so places with high concentrations of people hardly ever received the service. The municipal collection service disposes of waste in the municipal sanitary landfill (Figure 4), which is the only official site available to the municipality.

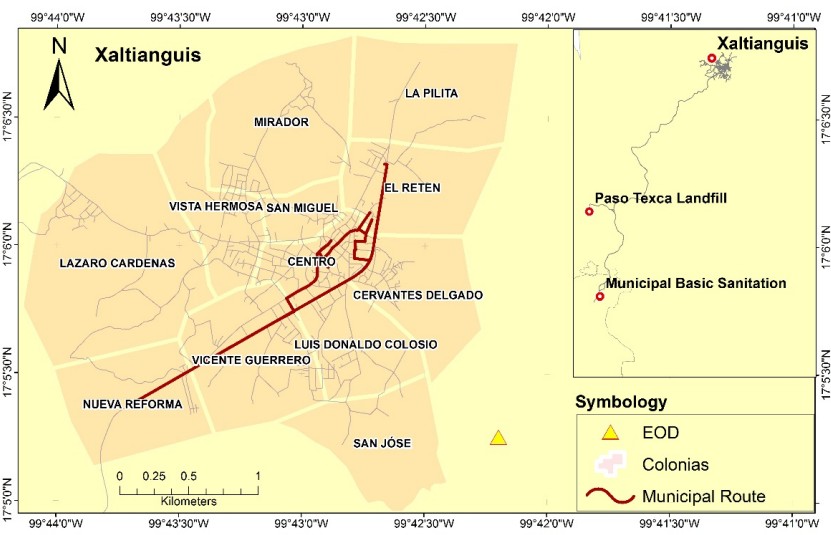

**Figure 4.** Current coverage of the municipal waste collection route.

The 2021 collection method was similar to that of 2018 and 2019, that is, sidewalks and fixed stops, but the collection frequency was sometimes twice a week, while in 2018 and 2019, it was once a week. It should be noted that such a frequency is complied with as long as the municipal collection vehicle does not present mechanical failures or is designated to the tourist area. In 2021, the collection was performed on Mondays and Thursdays, on a schedule from 8:00 to 17:00 h (9.0 h), with an average of 60 stops per day. This frequency may decrease for the aforementioned reasons, then the town remains unattended. The collection service for Xaltianguis is carried out by two trucks and two different drivers, one for each day. On Mondays, the following colonias are attended to: Nueva Reforma, Las Flores, Vicente Guerrero, Cervantes Delgado, El Retén, Luis Donaldo Colosio and Centro; on Thursdays, the following colonias are attended to: Nueva Reforma, Centro, San Miguel, Vista Hermosa, Vicente Guerrero and Luis Donaldo Colosio (Table 1). Thus, nine colonias are currently covered, while in 2019, only six of them were attended.

**Table 1.** Coverage of waste collection service by colonias using formal and informal routes.

| Colonia | Municipal Route | Informal Route 1 | Informal Route 2 | Informal Route 3 | Informal Route 4 |
|---|---|---|---|---|---|
| Centro | X | X | X | X | X |
| Lázaro Cárdenas | | | | | |
| El Reten | X | X | | X | |
| Vista Hermosa | X | | | X | |
| Vicente Guerrero | X | X | | X | |
| La Pilita | | | | | |
| Nueva Reforma | X | X | | X | |
| San Miguel | | | | X | |
| San José | | | | | |
| Luis Donaldo Colosio | X | X | X | | |
| Cervantes Delgado | | X | X | X | |
| El Mirador | | | | | |
| Total | 6 | 6 | 3 | 8 | 1 |

Note: X = Presence of the route in the colonia. Source: own elaboration with data obtained in the field.

### 3.2.2. Informal Service

The informal collection service (IS) in Xaltianguis in 2018 and 2019 consisted of four private vehicles, which were independent of each other (Table 2). Despite having particular logistics, they conducted improvised collection routes. This sector provided the service daily, prioritizing collection in the center of town, in contrast to the formal BSM collection, which was twice a week. With reference to the information presented in Table 2, informal routes 1, 2 and 3 carried out the intradomiciliary collection method (take and bring), with the exception of informal route 4, which only collected from the central market of the locality.

**Table 2.** Description and capacity of vehicles in the informal waste collection sector.

| Route Number | Vehicle Characteristics | Vehicle Dimensions (meters) | Volume of Waste (m³) | Quantity kg/vehicle |
|---|---|---|---|---|
| 1 | Ford Ranger XLT Mod. 92 | L 2.18 × W 1.50 × H 1.51 | 4.93 | 987.54 * |
| 2 | Mazda 2200 Mod. 88 | L 1.89 × W 1.59 × H 1.27 | 3.81 | 763.28 ** |
| 3 | Nissan Mod.93 | L 1.92 × W 1.54 × H 1.00 | 2.95 | 591.36 *** |
| 4 | Ford Ranger Mod. 94 | L 2.15 × W 1.42 × H 1.40 | 4.27 | 854.84 * |
| | Total MSW collected (kg/day): | | | 5143.02 |

Note: Mod. = model, L = length, W = width and H= height. * one trip/day, ** two trips/day, *** and three trips/day. Source: own elaboration with data obtained in the field.

The IS vehicles had their own logistics but did not ensure total coverage due to the fact that the colonias far from the center of the locality (Figure 5), such as Lázaro Cárdenas, La Pilita, El Mirador and San José, represent higher fuel consumption and cost and, therefore, less income, and some areas have streets that present an inaccessible topography. The service fee was different for each route, ranging from USD 0.50 to 1. The vehicles are non-conventional and without compaction, specifically the pick-up trucks (Figure 6). The collection frequency of informal routes 1, 2 and 3 was daily; only route 4 was twice a week. The informal collection routes covered only eight colonias, so four colonias did not have MSW collection; these were Lazaro Cardenas, El Mirador, La Pilita and San Jose (Table 1).

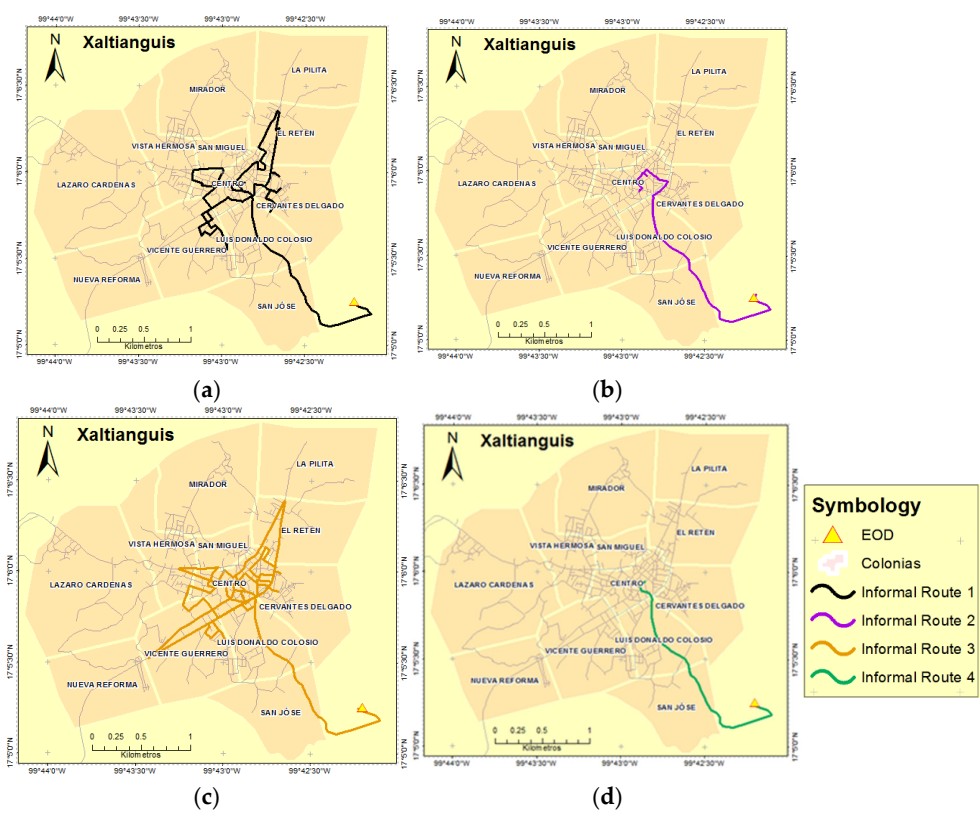

**Figure 5.** Coverage of informal waste collection routes. (**a**) Informal route 1, (**b**) informal route 2, (**c**) informal route 3, and (**d**) informal route 4.

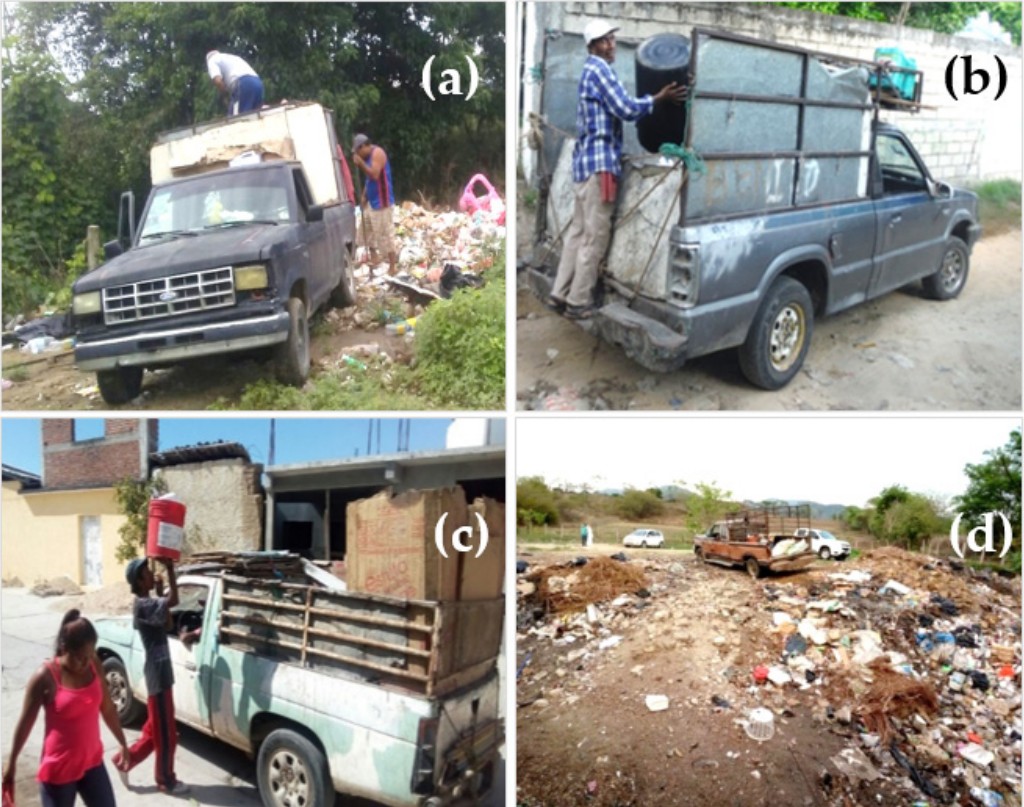

**Figure 6.** Informal service vehicles for waste collection. (**a**) Informal route 1, (**b**) informal route 2, (**c**) informal route 3 and (**d**) informal route 4.

The informal collection service currently consists of two pick-up trucks, with improvised routes and schedules; they provide services to colonias that have been served since 2019. Such routes were not geo-referenced to the present because the IS indicated not to modify the previous routes; these routes are shown in Figure 5.

Figure 6 shows the informal collection of Xaltianguis, carried out in 2018 and 2019, where four vehicles operated (a–d). In 2021, only two vehicles (a,b) continue to work due to the increased frequency of the collection of the BSM.

### 3.3. Route Optimization

Figure 7 presents a proposal for waste collection to optimize the municipal route service. The network analysis tool suggests an optimal scenario that consists of the design of a logical and efficient route that allows a series of criteria set out for the needs of the project to be met. In this case, time and distance were considered as impedance, in addition to the other criteria described in Table 3. The municipal route has little service coverage; it has a reduced frequency of home visits, so the service is affected.

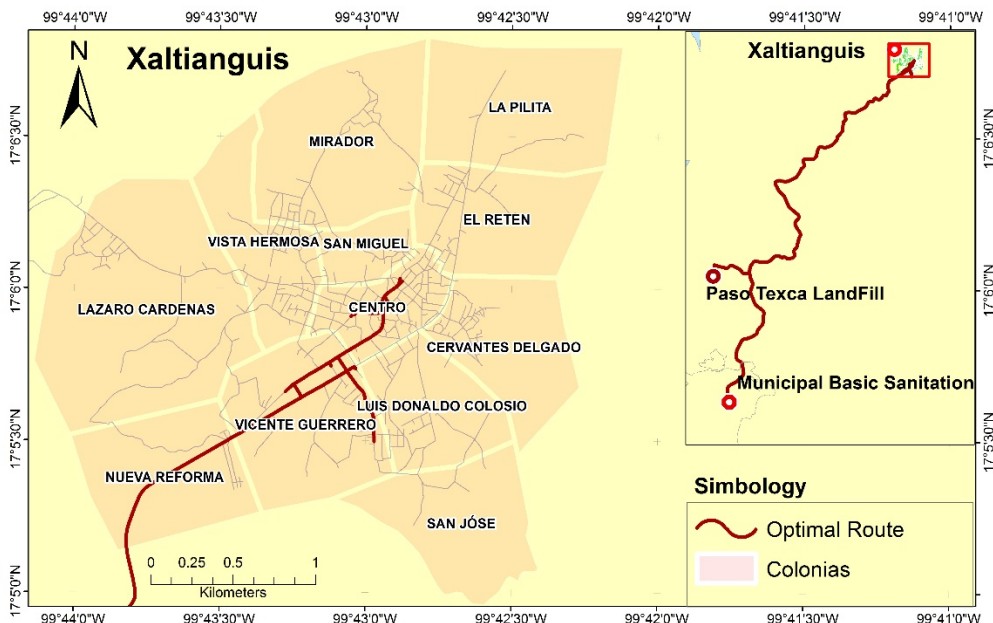

**Figure 7.** Optimization of the municipal route to the landfill. Source: own elaboration.

**Table 3.** Attributes of the optimized municipal route vs. current municipal route.

| Field | Municipal Route | Optimized Route |
|---|---|---|
| ObjectID | 1 | 1 |
| Name | Route 01 | Route 02 |
| Description | Waste Collection | Waste Collection |
| StartDepotName | Municipal Basic Sanitation | Municipal Basic Sanitation |
| EndDepotName | Paso Texca Landfill | Paso Texca Landfill |
| StartDepotServiceTime | 1 | 1 |
| EndDepotServiceTime | 1 | 1 |
| CostPerUnitTime | 1 | 1 |
| MaxOrderCount | 60 | 95 |
| OrderCount | 60 | 95 |
| TotalCost | 9.02 | 7.04 |
| RegularTimeCost | 9.02 | 7.04 |
| TotalTime | 9.02 | 7.04 |
| TotalOrderServiceTime | 4.15 | 4.75 |
| TotalDistance | 19,450.00 | 19,477.95 |

Source: own elaboration.

The trucks spend the night at the SBM's facilities, 32 km from the town. The trucks start at that point, where they will operate for an average of 7.04 h; this covers the workday in accordance with Mexico's current labor regulations. Once the truck reaches its load capacity, it is driven to the municipal sanitary landfill, which has the conditions for the treatment of MSW. Access to the facilities and the unloading maneuver takes about 45 min, which is considered in the configuration of the tool, and finally, they go to the BSM where all the units or trucks are placed.

## 4. Discussion

### 4.1. Technical and Logistical Conditions for Waste Collection

Regarding the technical and logistical conditions of waste collection in the locality, two types of service are described: the formal or municipal route service and the informal collection system. In this study, the formal route has areas of opportunity with the potential for improvement, among which the service coverage stands out. However, technical and logistical complications are also common, among which is the obsolescence of the collection vehicle fleet, which has not been replaced in a timely manner; in fact, in almost a third of the country, it is more than 20 years old [7].

As for the informal sector, it supports the municipal collection service, as in other places such as India (Sharholy et al. [30]) and China (Wang et al. [31]), where informal sectors positively influence MSWM and sustain its operation through variable tariffs just as formal routes operate in the provinces of Italy [32]. However, they are rarely within the legal framework, that is, they operate outside the law. Such is the case in Xaltianguis, where most of the collection is not recognized by municipal authorities. Xaltianguis is a town distant from the municipal capital (Acapulco, Mexico), where there is a collection service. However, it is normal for localities farther away from the municipal capital to lack the provision of different services [33].

In this regard, Salazar-Adams [8] points out that deficiencies in MSWM are an indicator of other deficiencies in local administrations. Therefore, a municipality incapable of carrying out effective MSWM will hardly be able to manage more complex public services such as education, transport, or sanitation, as collection is an essential service that a local authority is expected to offer to its inhabitants [8]. Although collection is part of MSWM, the operation and maintenance of the vehicles are considered a process that involves a higher cost compared to other stages of management [34]. Araiza and José [16] warn that poor design in micro- and macro-routing can result in serious damage to the collection system, deterioration of equipment, loss of time, reduced service coverage, increased costs, and even the proliferation of open dumps.

The optimization of the municipal route improved the service time by 2 h and had a greater number of stops (35 stops), while the distance had no improvement. Therefore, time is a determining factor to be considered as a feasible proposal. The study locality has road and geographic characteristics that affect optimization; a similar situation was reported by Mukama et al. [35] in Uganda, where MSW collection is currently one of the most critical public services because it is almost completely unprovided in marginal areas, and in the cases where it is provided, its coverage is low, which has caused public complaints. According to Oakley and Jimenez [36], another common practice is the burial of waste in community pits or the intentional burning of MSW in the open air. The same happens in Xaltianguis, where the inhabitants with a need to dispose of waste carry out these inadequate practices without taking advantage of recyclable materials.

### 4.2. Optimization of the Municipal Waste Collection Route

In this research, a spatial analysis of the MSW collection stage in Xaltianguis was carried out to formulate an adequate, relevant and viable proposal for waste management. Among the main challenges that municipalities face, inadequate budgets for basic services stand out [3], particularly in sanitation, which involves the collection, transport, and final disposal of MSW. The services of MSW can account for up to 50% of the municipal budget,

mainly this is related to the waste collection phase [5]. In MSWM issues, although the population plays an essential role in terms of collection, the municipality is responsible for the design and planning of the routes.

This is the case in the municipality of Acapulco de Juárez, where financial resources are limited to cover the collection service from 227 localities, among which Xaltianguis is included [22]. In this sense, Batista et al. [37] point out that few municipalities have an efficient MSWM service, and this situation has become a turning point for local development and a priority issue in tourist places. Two compactor vehicles have been assigned to the town, and they alternate their use. They make one trip per day and have the collection area sectioned for visits twice a week.

In the improvement of the efficiency of MSWM, it is necessary to reduce its associated costs with the purpose of providing a better quality of service and complying with the regulatory requirements established both globally and locally [38]. Although strategies in waste collection and the optimization of a municipal route are perceived as a viable alternative, Ferronato et al. [39] point out that the scientific literature lacks integrative and holistic (environmental, economic and management) evaluations of waste collection routes to increase the service coverage and reduce open dumping.

It has been suggested to use technologies that provide optimal scenarios to help solve the problem of waste. For this purpose, there are several pieces of GIS software for performing network analysis, but according to Cárdenas-Moreno et al. [27], ArcGIS is the most widely used software for waste management studies. In this sense, a study conducted in Ecuador showed that once the geo-routing proposal has been approved, it is imperative to carry out a socialization campaign among the inhabitants before implementing it; therefore, the commitment and work of the institutions related to MSWM and higher education institutions are fundamental [40].

In Xaltianguis, much of the waste collection is carried out informally, with personnel who work without salary or legal benefits provided by the municipality; these workers live on the fees contributed by the population, as well as on the recycling activity, a common practice carried out in other places to reduce poverty [41]. Future perspectives related to the improvement of MSWM must be focused in such a way that they adhere to sustainable practices to ensure a better urban living environment which will increase the economic productivity of the region, promote direct benefits for public health, and facilitate safe, dignified and legally protected employment opportunities [3]. It is crucial to note that urban conditions cause differences in distance or time reduction and can also influence modifications in weight, impedance or cost [21].

Network analysis can generate the shortest or fastest path of the selected impedance, and the path of lowest impedance or cost can be interpreted as the best option [28]. On this subject, Das et al. [20] point out that when distance is considered as impedance, in many cases, the shortest path in distance does not always imply a reduction in time. It is important to note that, in this study, steep and unpaved roads with difficult access were found, and this allowed the expansion of the information obtained from the GIS. These topographic conditions are part of the reality that generates a variation in the practice of solid waste management [3].

GIS-based tools allow optimizing routes under a spatial analysis [15–17,34,42]; however, this study involved the social aspect revealing its current formal and informal management, in addition to taking into account the conditions of the territory far from the municipal capital and without alternative routes; these characteristics that make it different from other studies. This research has a limited scope, determined by the objective, by proposing only the use of a tool such as GIS, but it can be improved when discussing other criteria (containers, transfer stations, etc.) related to the waste management system through mixed integer linear programming models (MILP) or deterministic and stochastic models to develop an optimization in the municipal solid waste sustainable management system [43,44], including models for an integrated supply chain network for MSW management [45].

*4.3. Practical Implications of the Study (Challenges, Strategies and Perspectives)*

The main challenges in this study are the lack of information from the responsible authorities on the volume of waste generated in the study locality, in addition to the limited financial, technical and logistical resources. Other challenges faced and described in the literature are the awareness of environmental pollution, social inclusion, technical knowledge and political will [46]. In this respect, Bernardes and Günther [33] point out serious repercussions in rural localities due to the lack of essential basic services, which can seriously affect the living conditions of the population. Such is the case of Xaltianguis, where, being far from the municipal capital, the collection service implies greater investment in the BSM's budget. According to field visits for this case, the collection implies a set of fixed stops which can last from two to four minutes each, and during the eight-hour workday, it covers an average of 95 stops; however, the number of stops per day is uneven.

Although an increasing trend in waste generation is expected in the coming years, it is necessary to integrate public policies that ensure the control of MSW from the source, strengthened with regulatory mechanisms and directed towards the objectives of sustainable development. According to Del Carmen et al. [29], the locality produces an average of 2.8 tons per day, and its largest volume is organic waste (68.3%), followed by non-recyclable inorganic (21.2%) and recyclable (10.50%) waste. In this regard, Banguera et al. [4] add that the generation of MSW is exceeding the capacity of being recycled and reused, thus turning a possible solution into an environmental problem. According to Saja et al. [3], the waste generated in a community is categorized as agricultural, mining, construction, and municipal solid. In this regard, it is recommended that integrated management that takes into account the environmental regulatory framework and addresses strict monitoring be implemented.

Therefore, a joint effort is sought with authorities, academia, and society to work on participation at the community level, which makes it possible to avoid social conflicts, improve the quality of life and develop sustainable practices in communities by proposing actions for the care of the environment and enabling the search for economic incentives to strengthen social relations in marginalized communities that require urgent attention on the issue of solid waste management. In this sense, Liu and Liao [47] point out that in practice, cooperation is required among recyclable waste collection and transportation companies as this allows them to improve the management of their resources, meet the needs of customers or citizens, and contribute to the improvement of environmental quality.

## 5. Conclusions

The optimization of the municipal waste collection route through the network analysis tool allowed us to provide a proposal to improve the service in this locality. While in the current schedule, nine hours are spent making 60 stops with the proposal, the travel time is reduced to seven hours. The analysis also helped to identify the need to increase the frequency of collection departures to provide greater coverage along the formal route, which is currently scheduled for two days a week. If the daily generation is 2.8 tons per day, a minimum frequency of three departures a week is required.

Regarding informal routes, it is concluded that they are necessary to complement the collection system. However, it is important that the authorities monitor this service to ensure that the collected waste actually reaches the place assigned for final disposal and is not sent to open dumps. Additionally, coordinated collaboration among government agencies, the informal sector and recycling companies can contribute to greater efficiency in both the solid waste reduction and collection stages, which can help to improve quality of life and lead to more environmentally responsible actions. From a social perspective, involving the locality in activities related to MSWM requires active community participation to carry out initiatives that provide solutions to MSW problems.

**Author Contributions:** Conceptualization, V.D.C.-N. and A.L.J.-L.; data curation, V.D.C.-N.; formal analysis, R.H.-N. and M.R.-U.; investigation, V.D.C.-N.; methodology, R.H.-N. and M.R.-U.; resources, V.D.C.-N., A.L.J.-L. and M.L.S.-R.; supervision, V.D.C.-N.; validation, A.L.J.-L., M.L.S.-R. and M.R.-U.; writing—original draft, V.D.C.-N. and R.H.-N.; writing—review and editing, V.D.C.-N., R.H.-N., A.L.J.-L., M.L.S.-R. and M.R.-U. All authors have read and agreed to the published version of the manuscript.

**Funding:** This research received no external funding.

**Institutional Review Board Statement:** Not applicable.

**Informed Consent Statement:** Informed consent was obtained from all subjects involved in the study.

**Data Availability Statement:** Not applicable.

**Conflicts of Interest:** The authors declare no conflict of interest.

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
