# Peer review of "Municipal Solid Waste Collection: Challenges, Strategies and Perspectives in the Optimization of a Municipal Route in a Southern Mexican Town"

_sustainability, doi:10.3390/su15021083_

Round 1

Reviewer 1 Report

The manuscript " Municipal Solid Waste Collection: Challenges, Strategies and Perspectives in the Optimization of a Municipal Route in a Southern Mexican town” analyzed the technical and logistical conditions of the collection and proposes the planning of an improved municipal route for waste collection using Geographic Information Systems. The manuscript is original and interesting, however there are some concerns and points which must be improved. Authors must work on the following points:

* The abstract should be revised. The authors should briefly discuss the purpose of the research and mention their findings adopted in this study. More quantitative data needs to be provided in the abstract.

*Provide significant words which are more relevant to the work in logical sequence as ‘keywords’.

* What is the current level of understanding in relation to the generation of Municipal solid waste and its management in Mexico? What are the knowledge gaps?. These should be included in the introduction section. The introduction is insufficient to provide the state of the art in the topic. Hypothesis should be given. How this work is different from the available data?

*The originality and novelty of the paper need to be further clarified. What progress against the most recent state-of-the-art similar studies was made in this study?

* For citations and reference within the text, author must follow guide for authors. The references must be also in the format of the journal.

* The discussion and interpretation of results does not clearly explain its impact on the literature and the field. Authors are suggested to add discussion by explaining trends in the obtained results along with the possible mechanisms behind the trends.

* It is strongly recommended to add a subsection, 'practical implications of this study,' outlining the challenges in the current research, future work, and recommendations, before the conclusion.

* Pls. conclude with more focus on the major outcomes of the paper.  

* The Table legends, figure captions and foot notes need improvement. All legends, captions and foot notes should have enough description for a reader to understand the table/figure without having to refer back to the main text of the manuscript.

* Check and correct grammatical and space errors throughout the article.

Author Response

Consulte el archivo adjunto.

Reviewer 2 Report

The article focuses on a specific and little-studied problem of waste collection optimization in a selected area of ​​southern Mexico.

To solve the problem posed in this way, the authors use some data, which, however, often lack the listed sources from which they were extracted (e.g. what is the source of the data about the volume of waste generated per day 2.7 tons, per week 18.9 tons? etc.). The Mexican laws, regulations and limits that the authors mention in the text are not cited. Great part of the data is only “estimated.”

In the keywords it is not necessary use capitals for all terms (e.g. Municipal Solid Waste Management).

The article is too long (21 pages). According to the opinion of the reviewer, 10-11 pages would be enough to explain the problem. Mainly the Introduction and the Discussion are very long. Their substantial shortening is necessary. In the Abstract don’t use abbreviations (SWM, MSWM). It is usual explain them only in the text. Also the part Conclusions is too long. The Conclusions should contain only the main results in a concise form.

In the table 2 is the total MSW collected per day (in kg) given only in case of Nissan Mod.93. Why?

Round 2

Reviewer 1 Report

The revised manuscript is in order and the current version can be considered for publication.

Author Response

Consulte el archivo adjunto.
